# Pilot Study: The Effects of Slice Parameters and the Interobserver Measurement Variability in Computed Tomographic Hepatic Volumetry in Dogs without Hepatic Disease

**DOI:** 10.3390/vetsci10030177

**Published:** 2023-02-22

**Authors:** Kosuke Kinoshita, Hitomi Kurihara, George E. Moore, Masahiro Murakami

**Affiliations:** 1Department of Veterinary Clinical Sciences, College of Veterinary Medicine, Purdue University, West Lafayette, IN 47907, USA; 2Department of Veterinary Administration, College of Veterinary Medicine, Purdue University, West Lafayette, IN 47907, USA

**Keywords:** computed tomography, hepatic volumetry, dogs, interobserver variability, slice thickness, slice number

## Abstract

**Simple Summary:**

Computed tomographic hepatic volumetry is usually time-consuming due to large numbers of slices. This study aimed to evaluate the relationship between slice interval and number of slices on hepatic volume in dogs using CT and to determine the interobserver variability of CT volumetric measurements for dogs without evidence of hepatobiliary disease which had undergone abdominal CT using either 2.5-mm or 3.75-mm slice thicknesses. Interobserver variability was low, with a mean percent difference in the hepatic volume of 3.3% among all observers. The mean percent differences in hepatic volumes generally increased when using larger slice intervals; however, the greatest percent difference was not always larger in larger slice intervals. The greatest percent differences in hepatic volume were decreased when using larger numbers of slices; the percent differences were <5% when using ≥20 slices for hepatic volumetry. Manual CT hepatic volumetry can be a good tool for a non-invasive assessment of liver volume with low interobserver variability in dogs, and a relatively reliable result would be acquired using ≥20 slices in dogs.

**Abstract:**

Manual computed tomographic (CT) hepatic volumetry is a non-invasive method for assessing liver volume. However, it is time-consuming with large numbers of slices. Reducing the slice number would expedite the process, but the effect of fewer slices on the accuracy of volumetric measurements in dogs has not been investigated. The objectives of this study were to evaluate the relationship between slice interval and the number of slices on hepatic volume in dogs using CT hepatic volumetry and the interobserver variability of CT volumetric measurements. We retrospectively reviewed medical records for dogs without evidence of hepatobiliary disease with abdominal CT from 2019 to 2020. Hepatic volumes were calculated by using all slices, and interobserver variability was calculated using the same dataset in 16 dogs by three observers. Interobserver variability was low, with a mean (±SD) percent difference in the hepatic volume of 3.3 (±2.5)% among all observers. The greatest percent differences in hepatic volume were decreased when using larger numbers of slices; the percent differences were <5% when using ≥20 slices for hepatic volumetry. Manual CT hepatic volumetry can be used in dogs to non-invasively assess liver volume with low interobserver variability, and a relatively reliable result can be acquired using ≥20 slices in dogs.

## 1. Introduction

Liver size has been reported to be one of the major diagnostic imaging criteria for the assessment of hepatic diseases in the human and veterinary medicine [1,2]. In dogs, radiography is routinely used to estimate the liver size to evaluate liver function in patients with suspected hepatic diseases [1]. Radiographic evaluation of liver size in dogs is difficult due to wide variation in patient sizes and body conformations, and relative measurement is used rather than absolute measurement. The commonly used internal controls to normalize patient variation for radiographic liver size evaluation in dogs include the vertebral body length and the gastric axis relative to the angle of rib or vertebrae [3,4,5]. However, the accuracy of radiographic diagnosis of hepatomegaly or microhepatia remains poor in dogs [3,4].

Manual computed tomographic (CT) hepatic volumetry is a non-invasive method for accurately measuring liver volume in humans and dogs [6]. CT-derived measurement of liver volume has been reported to correlate closely with the actual weight of liver explant regardless of the etiology of chronic liver disease in human patients [7]. Various studies demonstrated that CT-derived liver volume measurement had been used in humans for pre-operative planning of portal vein embolization and hepatectomy, as well as for post-operative assessment of functional reserve following liver transplantation [7,8,9].

Manual segmentation is considered the gold standard method of CT-derived liver volume measurement for pre-operative and post-operative hepatic volumetry in humans [8]. However, to perform manual CT hepatic volumetry, the manual drawing of the region of interest (ROI) for segmentation has to be performed on every slice of the CT study and is tedious and time-consuming when using large numbers of slices.

Increasing slice interval by reducing the number of manual drawing slices in performing manual CT hepatic volumetry would shorten this time-consuming process, but it will also increase the error of the result of manual CT hepatic volumetry. The maximum error of 5% in the calculated liver graft volume by CT volumetry is reported to be acceptable and will not have a significant clinical impact in the human medicine [10]. One study showed that 5-mm slice thickness could accomplish a maximum error of 5% in CT hepatic volumetry in humans [10]. However, the effect of fewer slices on the accuracy of CT hepatic volumetry in dogs has not been investigated. To accurately evaluate the effect of slice numbers on the accuracy of CT hepatic volumetry, the interobserver variability should be investigated to know the potential error of manual CT hepatic volumetry. The coefficient of variation (CV) is a common indicator of interobserver variability. Commonly, less than 10% of the CV is considered acceptable. The first objective of this study was to evaluate the interobserver variability of manual CT hepatic volumetry among three observers using CV. The second objective was to investigate which slice intervals and number of slices used for CT hepatic volumetry would be able to maintain the greatest percent differences of less than 5% of regular CT hepatic volumetry using every slice.

## 2. Materials and Methods

### 2.1. Patient Selection

This is a retrospective case series study. Purdue University Veterinary Teaching Hospital (PUVTH) Medical Record database, from 18 December 2019 to 24 July 2020, was searched retrospectively to identify dogs that had undergone abdominal CT, using either 2.5 mm or 3.75 mm slice thicknesses. Abdominal CT reports, blood work results, and medical records from the same presentation were reviewed by the board-certified radiologist (M.M.). Dogs with abnormal CT findings of the liver, abnormal blood work (complete blood count and serum enzyme level including alanine aminotransferase, alkaline phosphatase, gamma-glutamyl transferase, total bilirubin, albumin, and glucose), and history or clinical signs of hepatic diseases were excluded. Dogs with a history of continuous administration of steroids were also excluded. The retrospective search was continued until eight dogs who met the above criteria were collected in each slice thickness (2.5-mm or 3.75-mm). Sixteen dogs met inclusion criteria, including the mixed-breed dog (3), Labrador Retriever (2), Golden Retriever (2), Jack Russel Terrier (1), German Shorthaired Pointer (1), Belgian Malinois (1), Goldendoodle (1), Bulldog (1), Rottweiler (1), Alaskan Malamute (1), German Shepherd (1), and Mastiff (1). They were 5 sprayed females, 9 castrated males, and 2 intact males. The ages of the included dogs were 6.4 ± 3.5 years old (mean ± SD). Body weights were 32.6 ± 9.8 kg (mean ± SD) (Table 1).

### 2.2. Manual CT Hepatic Volumetry

All CT studies were acquired with the dogs under general anesthesia or heavy sedation in sternal recumbency using the 64-slice third-generation CT units (Light Speed VCT, GE Medical Systems Inc., Waukesha, WI, USA) in helical scan mode. Pre- and post-contrast studies of the liver were reviewed by a board-certified radiologist (M.M.), and dogs with any hepatic abnormalities in the CT study were excluded from the present study. After exclusion, only pre-contrast studies were used for further CT hepatic volumetry. Manual CT hepatic volumetry was performed by a board-certified radiologist (M.M.) and two veterinarians (K.K. and H.M.) after training, supervised by a board-certified radiologist (M.M.), using a DICOM viewer (Horos 64-bit, Purview, Annapolis, MD, version 3.3.6.). The window width was set at 350 HU and the window level at 40 HU. Segmentation of the liver was performed by the manual drawing of the operator-defined region of interest (ROI) on pre-contrast transverse images of the entire liver from the cranial margin of the liver at the diaphragm to the most caudal margins of the liver. The hepatic vessels within liver parenchyma were included within the ROIs, and the gallbladder and visible hepatic lobe fissures and vessels present outside of the hepatic parenchymal margination (including caudal vena cava) were excluded (Figure 1). After manual drawing of ROIs on hepatic parenchyma in all transverse images, hepatic volume was calculated using the following formula: *Σ{each slice area (cm^2^)* × *slice thickness (cm)}.*

### 2.3. Interobserver Variability on Manual CT Hepatic Volumetry

Three observers (Μ. Μ, Μ. H, and K. K) independently performed manual CT hepatic volumetry in all 16 dogs. Full hepatic volume was calculated using all slices with no inter-slice gaps (2.5-full and 3.75-full groups). The following evaluations of interobserver variability were calculated using these full hepatic volume results by three observers. The Coefficient of Variation (CV) was calculated among hepatic volumes calculated by three observers in each patient using the following formula: CV = standard deviation/mean × 100. Then, the mean and standard deviation of CV was calculated in all patients as interobserver variability. The greatest difference in hepatic volume among the 3 observers was calculated for each patient (e.g., the difference between the largest volume and the smallest volume in each patient). Then, the greatest percent difference was calculated using the following formula: greatest percent difference = the greatest difference/average of hepatic volumes from three observers in each patient. The mean and standard deviations of the greatest percent difference in hepatic volume were calculated and recorded to describe the percent difference of hepatic volume measurements among three observers. Percent differences in hepatic volume measured between two observers were also calculated: between observer A and B, between observer B and C, and between observer A and C.

### 2.4. Effect of Different Slice Intervals on Manual CT Hepatic Volumetry

The effects of slice interval on volumetric measurements were evaluated. Hepatic volumetry with different slice intervals was calculated. Each group of hepatic volumetry performed with different slice intervals was named as the “slice thickness–slice intervals” group (e.g., the 2.5–5.0 group was the group using a 2.5 mm slice thickness and slice interval of 5.0 mm). In the 2.5–5.0 group, 2 hepatic volumes were calculated from 1 study using a different slice as the first slice (odd number of slices used for the first set and even number of slices used for the second set). Thus, 16 measurements were acquired in the 2.5–5.0 group (2 measurements in 8 dogs), as well as *n* = 24 in the 2.5–7.5 group (3 measurements in 8 dogs), *n* = 32 in the 2.5–10 group (4 measurements in 8 dogs), *n* = 48 in the 2.5–15 group (6 measurements in 8 dogs), *n* = 16 in the 3.75–7.5 group (2 measurements in 8 dogs), *n* = 24 in the 3.75–11.25 group (3 measurements in 8 dogs), and *n* = 32 in the 3.75–15 group (4 measurements in 8 dogs). Using manually drawn ROIs on hepatic parenchyma in each selected transverse images, the hepatic volume of CT images for different slice interval was calculated using the following modified formula: *Σ{each slice area (cm^2^)* × *slice thickness (cm)}* × *total number of slices of hepatic parenchyma/number of slices used for calculation*.

The percent difference between calculated hepatic volumes using various slice intervals and the corresponding full hepatic volumes were calculated using the following formula: *|(calculated hepatic volume of each slice interval) − (the corresponding full hepatic volume)|/(the corresponding full hepatic volume)* × 100. Then, the mean and the standard deviation of the percent difference of hepatic volume of each slice interval were calculated to evaluate the effects of slice intervals on volumetric measurements. The greatest percent differences of hepatic volume at each slice interval (e.g., for 2.5–7.5 and 3.75–7.5 groups had a 7.5 mm slice interval) was also calculated as the largest absolute difference of hepatic volume from the full hepatic volume at each slice interval divided by the full hepatic volume times 100.

### 2.5. Effect of Different Slice Numbers used on Manual CT Hepatic Volumetry

The effects of the number of slices used in hepatic volumetry were also evaluated. For all calculated hepatic volumes in different slice intervals (*n* = 576; 192 measurements in 3 observers), the numbers of slices used for each calculated hepatic volume was recorded. The relationship between the percent difference in hepatic volume from the corresponding full hepatic volume and the numbers of slices used for hepatic volumetry was evaluated. The greatest percent differences in hepatic volume compared to corresponding full hepatic volume for each number of slice used for hepatic volumetry were recorded for each group with different numbers of slices used for hepatic volumetry: numbers of slices used for hepatic volumetry, less than 10, 10 to 14, 15 to 19, 20 to 24, and more than 25 slices.

## 3. Results

### 3.1. Interobserver Variability

CT hepatic volumetry of dogs was performed in eight dogs using a 2.5 mm slice thickness and eight dogs using a 3.75 mm slice thickness. CT hepatic volumetry was performed in each dog by the three observers (*n* = 48; 16 dogs by 3 observers). The mean hepatic volume (±SD) was 722.2 (±281.2) cm^3^. The calculated full hepatic volumes of each dog by each observer are shown in Figure 2. The mean (±SD) CV of the hepatic volume calculated by three observers was 2.5 (±1.4)%. The mean (±SD) of the greatest percent difference among the three observers was 4.9 (±2.7)%: 2.8 (±2.0)% between observers A and B; 2.8 (±1.9)% between observers B and C; and 1.3 (±0.8)% between observers A and C.

### 3.2. Accuracy of Hepatic Volumetry in Each Slice Interval

Percent differences in the hepatic volume of each slice interval compared to the corresponding full hepatic volumes (2.5-full or 3.75-full) were shown in Figure 3. The means (±SD) of the percent difference in hepatic volume compared to the corresponding full hepatic volumes (2.5-full or 3.75-full) were 1.4 (±1.1)% in 2.5–5.0 group, 2.6 (±1.7)% in 2.5–7.5 group, 3.0 (±2.1)% in 2.5–10 group, 5.0 (±3.1)% in 2.5–15 group, 1.3 (±1.2)% in 3.75–7.5 group, 2.4 (±1.5)% in 3.75–11.25 group, and 3.5 (±2.2)% in 3.75–15 group. The mean percent differences in hepatic volumes generally increased when using a larger slice interval (Figure 3). The greatest percent differences in hepatic volume compared to the corresponding full hepatic volume (2.5-full or 3.75-full group) were 3.6% at 5.0 mm slice interval (2.5–5.0 group), 3.8% at 7.5 mm slice interval (2.5–7.5 and 3.75–7.5 groups), 8.1% at 10 mm slice interval (2.5–10 group), 5.9% at 11.25 mm slice interval (3.75–11.25 group), and 6.7% at 15 mm slice interval (2.5–15 and 3.75–15 groups). The greatest percent difference in hepatic volumes was largest at 10 mm slice interval, larger than 11.25 and 15 mm slice intervals (Figure 3).

Percent differences in hepatic volume compared to the corresponding full hepatic volume (2.5-full or 3.75-full group) with numbers of slices used for hepatic volumetry are shown in Figure 4. The greatest percent differences in hepatic volume compared to the corresponding full hepatic volume (2.5-full or 3.75-full group) were 17.5% using less than 10 slices, 9.0% using 10 to 14 slices, 5.5% using 15 to 19 slices, 4.8% using 20 to 24 slices, and 2.6% using more than 25 slices (Figure 4).

## 4. Discussion

This study aimed to discuss the relationships between slice interval and the number of slices in calculating hepatic volume on CT to reduce the clinician’s burdens on manual tracing. CT liver volumetry technique has been used to estimate the volume of the liver accurately in vivo in human medicine and veterinary medicine [11,12,13,14]. The liver volume calculated using CT hepatic volumetry showed a correlation with the actual liver volume in human liver transplant patients, and CT hepatic volumetry to calculate liver volume was found to be an acceptable method [7]. Manual segmentation of ROIs is considered to be more accurate than automatic segmentation, but it is time-consuming with large numbers of slices. Thus, in the current study, the interobserver variability and the effect of slice intervals and slice numbers on the accuracy of CT hepatic volumetry were investigated.

Interobserver variability was defined as the difference in the measurements between observers, which is important to know for the accuracy of the examination technique [14,15]. Interobserver variability for manual CT volumetry in dogs has not been investigated. The coefficient of variation (CV) is the ratio of the standard deviation to the mean and can be an indicator of the interobserver variability [16]. A higher value of CV indicates large interobserver variability. Several veterinary publications have used CV to evaluate interobserver variability for measurement of the structures on the radiograph, such as cardiac size [17,18], tibial plateau angle [19], the severity of osteoarthritis [20], and size of pulmonary nodule [21]. Although no specific CV value is considered a small enough interobserver variability in those publications, commonly less than 10% of CV is considered very good. In the previous publication evaluating interobserver variability of pulmonary nodule diameter measurement, a CV of 16% was considered a low interobserver variability [21]. A previous manual CT hepatic volumetry publication used an intraclass correlation coefficient to evaluate interobserver variability, and an intraclass correlation coefficient of 0.957 with 95% confidence interval of 0.908–0.982 was considered excellent [22]. In the present study, CV of CT hepatic volumetry among three observers was 2.5 (±1.4)%. The mean of the greatest percent difference between the two observers was less than 5% in all three observers. Thus, in the present study, the interobserver variability of manual CT hepatic volumetry was considered very good. In our study, there was a difference in the experience of each observer since one of the observers was a board-certified veterinary radiologist, and the rest of the observers were veterinarians who were trained to perform CT hepatic volumetry for this study. Our results suggest that the CT volumetry method performed in this study was not influenced by the experience of each observer.

Hepatic volumetry in humans was frequently time-consuming due to manual drawing ROIs on large numbers of slices. Reducing the number of manual drawing slices can dramatically contribute to reducing the time of the CT hepatic volumetry procedure. On the other hand, the effect of using different numbers of slices in CT hepatic volumetry has been described, and smaller numbers of slices used for CT hepatic volumetry are known to show significantly different liver volume compared to actual liver volumes in humans [23]. Another study showed that CT hepatic volumetry calculated using a 2.5 mm slice thickness or thicker significantly underestimated liver volumes compared to the liver volume calculated from 3D images in humans [10]. Thus, it is crucial to know the minimum number of slices required for accurate manual CT hepatic volumetry in dogs. In some studies, less than a 5 to 10% error between CT volumetric liver volume and true liver graft volume was considered acceptable in transplantation medicine in humans [7,10]. Thus, in the present study, an error of less than 5% volume from CT volumetric liver volume using full slices (full hepatic volume) was considered to be acceptable, which was similar to the interobserver variability described in the previous study.

In the present study, the effect of different slice intervals used for manual CT hepatic volumetry was evaluated. The mean percent differences in hepatic volumes compared to the hepatic volume calculated using full slices were equal or more than 5% only in the 2.5–15 group, which used one slice in every six slices for hepatic volumetry. The mean percent differences in hepatic volumes generally increased when using a larger slice interval. However, the greatest percent difference was not always larger with using a larger slice interval, and the largest greatest percent difference (8.1%) was seen in the 2.5–10 group, which was using a 10 mm slice interval. The greatest percent differences in hepatic volume calculated using 11.25 mm (5.9%) and 15 mm (6.7%) slice interval groups were smaller than the 10 mm slice interval group (8.1%). One of the reasons why the greatest differences were not proportional to the slice intervals is thought to be caused by the difference in the number of slices used for manual CT hepatic volumetry.

According to the result of comparing different slice intervals, further evaluation was performed with a different number of slices used for manual CT hepatic volumetry. The greatest differences in calculated hepatic volumes using manual CT hepatic volumetry were proportionally decreasing with using the larger number of slices (Figure 4). The differences in hepatic volumes were always less than 5% if using equal or more than 20 slices. The number of slices containing liver parenchyma in the canine CT study is usually more than 40. In the present study, the numbers of slices containing the liver were 36 to 63. Reducing the number of slices for manual CT volumetry will help the reduction of process time and increase the utilization of this method.

There has been a publication on the effect of slice numbers used for CT hepatic volumetry in dogs using sagittal reformatted images [22]. This study suggested using more than nine slices of sagittal reformatted images to accurately perform CT hepatic volumetry [22]. However, in the previous study, the calculated hepatic volumes using a different number of slices (9 to 58 through 65 slices depending on the size of the dogs) showed wide variations in each dog with at least more than 10% volume difference from the mean calculated hepatic volume in all five dogs, and the differences of the largest volume and the smallest volume using different slice numbers in each dog were 11.1% to 37.9% [22]. In the present study, we evaluated the greatest difference in calculated hepatic volume compared to the full study and used 5% as a cut-off to reduce the error caused by using a smaller number of slices.

The size of dogs in the present study was limited. In the present study, the body weight of dogs used for CT hepatic volume ranged from 13.5 kg to 44.5 kg. Thus, further evaluation using dogs weighing less than 13.5 kg or more than 44.5 kg may need to verify the accuracy of using reduced slice numbers in CT hepatic volumetry in small or very large dogs.

In conclusion, manual CT hepatic volumetry is a feasible method of calculating hepatic volume with low interobserver variability, and the time of the procedure can be reduced by retaining the accuracy of using 20 or more slices.

## Figures and Tables

**Figure 1 vetsci-10-00177-f001:**
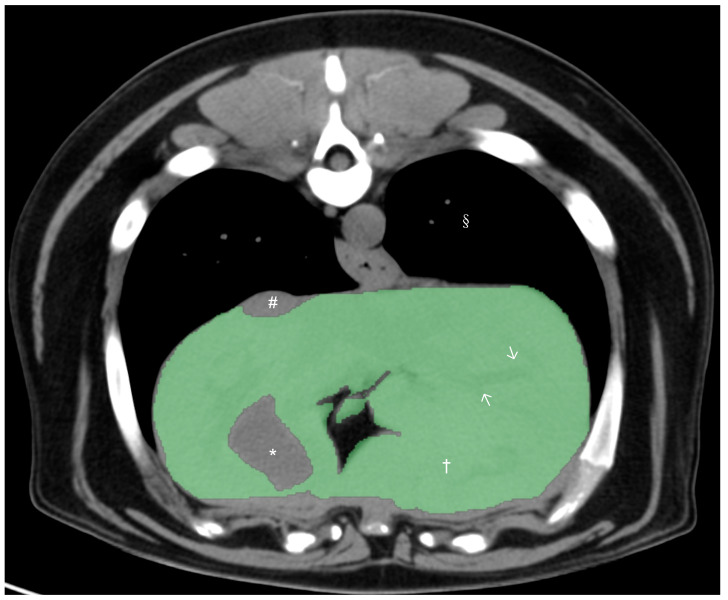
Precontrast transverse abdominal CT images used for measuring liver volume in dogs. The segmentation of the liver was manually selected as Region of Interests (ROIs: green regions). Note that the hepatic vessels within liver parenchyma were included (white arrows). The gall bladder, hepatic lobe fissure, and hepatic vessels present outside hepatic parenchyma were excluded. Window width, 350 HU, and window level, 40 HU. Liver parenchyma (†), gallbladder (*), caudal vena cava (#), and pulmonary parenchyma (§).

**Figure 2 vetsci-10-00177-f002:**
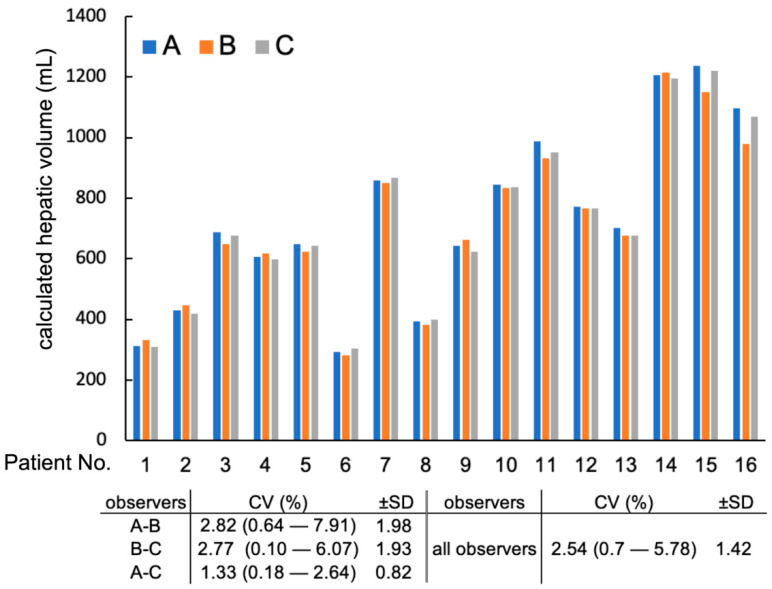
Interobserver variability of calculated full hepatic volumes by each observer. CV: Coefficient of Variation. Patients from 1 to 8 were with 2.5 mm slice intervals, whereas patients 9 to 16 were with 3.75 mm slice intervals.

**Figure 3 vetsci-10-00177-f003:**
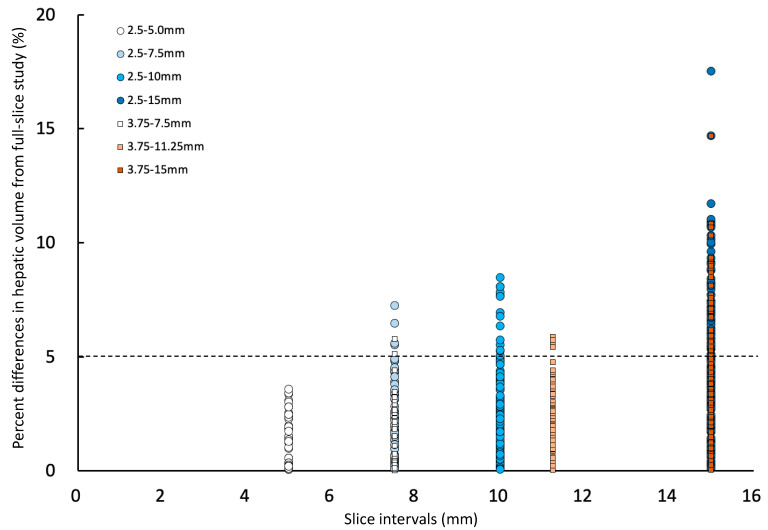
Percent differences in hepatic volume of each interslice gaps compared to 2.5 mm-full or 3.75 mm-full. The percent difference lower than dotted line (less than 5%) is considered acceptable.

**Figure 4 vetsci-10-00177-f004:**
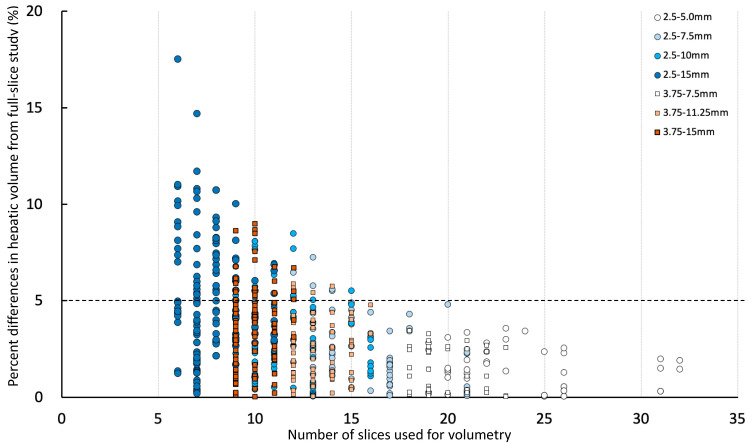
Percent differences in hepatic volume compared to 2.5-full or 3.75-full with numbers of slices used for hepatic volumetry. The percent difference lower than dotted line (less than 5%) is considered acceptable.

**Table 1 vetsci-10-00177-t001:** Summary of clinical information of dogs without hepatic diseases. Patients from 1 to 8 were with 2.5 mm slice intervals, whereas patients 9 to 16 were with 3.75 mm slice intervals.

	Case No.	Breed	BW	Ages	Sex
2.5 mm slice interval	Case 1	Jack Russel Terrier	22.05	10	FS
	Case 2	Labrador Retriever	44.53	13	MN
	Case 3	MBD	29.3	8	Male
	Case 4	German Shorthaired Pointer	25	6	MN
	Case 5	Belgie Malinois	27	5	Male
	Case 6	Goldendoodle	15.8	1	FS
	Case 7	MBD	33.4	8	FS
	Case 8	Bulldog	13.5	10	MN
3.75 mm slice interval	Case 9	Golden Retriever	32.5	0	MN
	Case 10	Golden Retriever	43	9	MN
	Case 11	Rottweire	34.7	4	FS
	Case 12	Alaskan Malamute	42.1	9	MN
	Case 13	German Shepherd Dogs	34.9	5	MN
	Case 14	Mastif	40	6	FS
	Case 15	MBD	38.6	2	MN
	Case 16	Labrador Retriever	44.5	7	MN

## Data Availability

The data not presented in the manuscript are available for consultation after a reasonable request to the corresponding authors.

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
