# Peer review of "Pilot Study: The Effects of Slice Parameters and the Interobserver Measurement Variability in Computed Tomographic Hepatic Volumetry in Dogs without Hepatic Disease"

_vetsci, 2023, doi:10.3390/vetsci10030177_

Round 1

Reviewer 1 Report

General comments: This is a study that evaluates the technique for calculating the volume of the liver in dogs using a CT scan to make a tedious process quicker. I’m finding it difficult to find the value in this as calculating liver volume is so rarely performed in veterinary medicine. In human medicine, this seems to be valuable for transplants; however, this is yet to be a standard of practice in our canine population. Additionally, in humans, there is likely a standard liver size for males/females that a calculated volume could be compared to. With our patients there is such a huge size range, that there is unlikely to ever be a standard that a measured volume could be compared to to objectively diagnose hepatomegaly or microhepatia. As mentioned in the introduction, this has been historically performed with radiographs using patient internal controls to account for size differences; however, here, you are simply calculating a volume which provides very little valuable information. While this is a well written, clear study, and you certainly achieve your objectives, there needs to be significant evidence for the practice of measuring the liver in dogs and what value it will add for this to be a recommended routine practice for radiologists evaluating CT images.

Title: Include that these are healthy dogs without liver disease

48: Reference

49: Reference

51: Reference

59: human”s”

86: Why 8?

87: Were dogs on prednisone excluded?

209: This should be determined relative to the patient size. For instance, 9 slices in a 3kg Chihuahua is probably pretty accurate; however, 9 slices in a 85kg Great Dane would have a high level of variability.

244: Should say “veterinarians who were trained”

255: remove “found”

296: At the authors’ institution, how frequently is the liver volume calculated? How much time will this be saving in the day to day operations?

Figure 2: Are patients 1-8 the 2.5mm slices and 9-16 the 3.75?

Reviewer 2 Report

The paper submitted by the authors is not well organized, showing important flaws in most of the sections. Therefore, I recommend rejection and encourage resubmission.   In the introduction section, I miss more information about the process. A retrospective study with only eight animals. The anatomical landmarks should be more precise. Please include labellings in Figure 1. 8 or 16 dogs, please clarify. In section 2.4. Number and groups are confused. The statistic treatment is not clear. What treatment did you use? How many slices did you get in each Group? Figs 2,3 are not cited in the text. The entire section (MM) has so many flaws and information that can be misinterpreted. Patient characteristics should be in material and Methods section (I recommend you include a table with this info). The first paragraph of the discussion section should be revised and rewritten.

Round 2

Reviewer 1 Report

Thank you for addressing my concerns. With these changes, I feel that the manuscript should be accepted for publication. 

Author Response

I really appreciate your revision.

Reviewer 2 Report

The revised paper provided by the authors does not fulfill the corrections suggested by the reviewer. Therefore, I recommend rejection.

Specific comments:

- The last paragraph of the introduction should be rewritten, as well as the first one of the material and methods section.

- The anatomical levels are not well described. Therefore, you should describe the segmentation from the diaphragmatic surface of the liver at the level of caudal vena cava sulcus…

- In my opinion is impossible to do a retrospective study with 16 animals. In a a Veterinary Hospital as yours,  I am sure that you perform at least more than 100 studies per month. Therefore, this number is very low. Moreover, it is important to highlight that these animals are not exotic animals, which could explain that low number.

- Again, the animals used should be in the material and methods section since age, breeds or sex are not results, it would be correct here when  you are adding a disease to those breeds 

- you can not explain statistic parameters without explaining the treatment used to analyze these parameters 
